# Urinary Incontinence in Young Gymnastics Athletes: A Scoping Review

**DOI:** 10.3390/sports13090319

**Published:** 2025-09-10

**Authors:** Alice Higounenc, Alice Carvalhais, Ágata Vieira, Sofia Lopes

**Affiliations:** 1Departamento de Tecnologias de Diagnóstico e Terapêutica, Escola Superior de Tecnologias da Saúde do Tâmega e Sousa, Instituto Politécnico de Saúde do Norte (IPSN), 4585-116 Gandra, Portugal; alice.higounenc@gmail.com (A.H.); alice.carvalhais@ipsn.cespt.pt (A.C.); agata.vieira@ipsn.cespt.pt (Á.V.); 2Instituto de Ciência e Inovação em Engenharia Mecânica e Engenharia Industrial (INEGI), 4200-465 Porto, Portugal; 3Departamento de Fisioterapia, Escola Superior Saúde Santa Maria, Trav. Antero de Quental 173/175, 4049-024 Porto, Portugal; 4H^2^M—Unidade de Investigação em Saúde e Movimento Humano, Instituto Politécnico de Saúde do Norte, CESPU, CRL, 4760-409 Vila Nova de Famalicão, Portugal; 5Centro de Investigação em Reabilitação (CIR), Escola Superior de Saúde, Instituto Politécnico do Porto, Rua Dr. António Bernardino de Almeida, 400, 4200-072 Porto, Portugal; 6ESS, Escola Superior de Saúde, Instituto Politécnico do Porto, Rua Dr. António Bernardino de Almeida, 400, 4200-072 Porto, Portugal

**Keywords:** athletes, exercise, gymnastic, pelvic floor dysfunction, pelvic floor muscles

## Abstract

Background/Objectives: Urinary incontinence (UI) is increasingly reported among young nulliparous women, especially those engaged in high-impact sports. This may increase the risk of developing stress urinary incontinence (SUI) later in life. This scoping review aims to synthesise current evidence on UI in gymnasts, identify the gymnastics modalities with the highest prevalence of UI, and examine the outcomes measures and interventions considered for UI. Methods: A scoping review was conducted using the following databases: PubMed, Cochrane, Science Direct, Scielo, EBSCO, PEDro, and NIH. Studies published in English or Portuguese between January 2012 and May 2023 were included. Review articles, qualitative studies, and conferences abstracts were excluded. Studies were analysed according to the PRISMA-ScR framework. Results: Out of 169 identified studies, 4 were included. SUI emerged as the most prevalent type of UI, particularly in artistic and trampoline modalities. All the studies used the ICIQ-UI-SF to assess UI. Reported outcomes included prevalence of UI and SUI, impact on quality of life and athletic performance, as well as knowledge about UI and pelvic floor. No studies investigated treatments for UI. Conclusions: The prevalence of SUI is high among gymnasts (70%), especially in artistic and trampoline disciplines. Although there is outcome heterogeneity across studies, all used the ICIQ-UI-SF. There is a critical gap regarding interventions for UI in this population.

## 1. Introduction

Urinary incontinence (UI) was defined by the International Continence Society (ICS) in 2009 as the involuntary loss of urine [1]. Different types of UI have been described, the most prevalent being stress urinary incontinence (SUI), urge urinary incontinence (UUI), and mixed urinary incontinence (MUI). SUI is defined as the complaint of involuntary loss of urine with exertion or physical activity, including sporting activities or the act of sneezing or coughing [2]. In this case, the intravesical pressure exceeds the maximum urethral pressure in the absence of contraction of the detrusor muscle. UUI is associated with an urgent urge to urinate, where there are involuntary contractions of the detrusor muscle, which is responsible for the symptoms. Mixed urinary incontinence (MUI) combines the two types mentioned above, i.e., leakage associated with urgency and increased PFM (IAP) [3]. The condition occurs in both sexes, but it is more common in women [4]. Urinary incontinence has various repercussions for daily activities, social interactions, and women’s perception of their own health. Documented outcomes include poor social and mental wellbeing, social isolation, low self-esteem, and depression, which significantly affect quality of life and may lead to psychological, physical, professional, sexual, and social consequences [3]. In a systematic review of worldwide studies, the prevalence of UI ranges from 5% to 70%, with most studies reporting a prevalence of any type of UI of between 25% and 45% [5]. An epidemiological study carried out in the Portuguese population in adults over 40 in 2008 by the Faculty of Medicine of the University of Porto in conjunction with the Portuguese Association of Urology (APU) and the Portuguese Association of Neuro-Urogynecology (APNUG) found a prevalence of UI of 15.1% in the population, 21.4% in women and 7.6% in men [6,7]. This prevalence tends to increase with age [8]. However, this problem also affects young, physically active women, even in the absence of risk factors [9]. In fact, studies show an increase in the prevalence of UI in young, nulliparous women [10,11], especially when they practice high-impact sports [11,12,13,14]. The more frequent the impact associated with increased intra-abdominal pressure, the greater the need for containment and support of the pelvic organs by the pelvic floor muscles, which must be trained to preserve their function [15,16]. Prevalence varies not only with impact, but also with exercise intensity. In fact, scientific evidence shows that high-impact sports and participation in long-term competitions are associated with a higher prevalence of SUI in nulliparous women, especially trampolinists [15,17,18,19]. However, the majority of athletes leak urine during training and not during competitions (95.2% vs. 51.2%, respectively). This may be due to higher levels of catecholamines (norepinephrine and epinephrine) released from the sympathetic nervous system and adrenal medulla during stressful situations, which bind to urethral alpha receptors to maintain urethral closure during stressful competitions [18,20]. This response to stressors can be modified by the characteristics of the stressor stimulus, and the ratio of epinephrine to norepinephrine varies depending on the nature of the stress. Mental stress mainly increases the level of epinephrine, while the combination of physical and mental exertion stimulates the production of norepinephrine, the main agonist of urethral alpha receptors [21,22]. Therefore, and based on what has been described, the aim of this scoping review was to summarise current evidence on the prevalence of UI among adolescent and adult female gymnasts, to identify which gymnastics disciplines have the highest prevalence of UI, and to describe the assessment instruments, reported outcomes, and any interventions studied. The following research questions guided this review: (1) Which gymnastics disciplines have the highest prevalence of UI in female athletes? (2) What instruments and outcomes are used to assess UI in this population, and what interventions have been reported?

## 2. Materials and Methods

This scoping review was carried out in accordance with the guidelines to the structure of Preferred Reporting Items for Systematic Reviews and Meta-Analysis extension for Scoping Reviews (PRISMA-ScR) [23] and the methodology proposed by Joanna Briggs Institute (JBI) Manual for evidence synthesis (2020) [24]. Registration was completed on Open Science Framework (OSF).

### 2.1. Eligibility Criteria

The eligibility criteria were established using the acronym PCC (Population, Concept, and Context) in accordance with the JBI methodology [24]. 

Population: adolescent and adult female gymnasts.Concept: evaluation of stress urinary incontinence in gymnasts.Context: to assess incontinence in a training environment, in a sports context.

### 2.2. Evidence Sources

Primary and secondary studies were considered in this scoping review. Review studies (literature and systematic), qualitative studies and abstracts of papers presented at congresses/conferences were excluded. Studies published in English and Portuguese were considered. Only studies published from 1 January 2012 to 31 May 2023 were included. The search for this information was carried out between April and May 2023.

### 2.3. Research Strategy

The search was carried out in the following databases: PubMed, Cochrane, Science Direct, Scielo, EBSCO, PEDro, NIH (ClinicalTrials.gov) based on the following research strategy: (“woman” OR “female” OR “athlete”) AND (“gymnastic” OR “trampoline” OR “acrobatic” OR “high impact sport”) AND (“urinary incontinence” OR “stress urinary incontinence” OR “pelvic floor disorders” OR “loss of urine” OR “urine leakage”) AND (“prevalence” OR “treatment” OR “knowledge” OR “impact” OR “quality of life” OR “prevention”). Depending on each database, adaptations were made to the research expression, as can be seen in Table 1.

### 2.4. Evidence Selection

The collection was carried out by just one reviewer (AH). After the initial database searches, grey literature was searched using Google Scholar. After the final selection, the principal investigator independently extracted the information from the articles eligible for data synthesis. The 2nd researcher (SL) was always present throughout the process, also analysing independently and together whenever there were doubts about whether to include the articles. The most relevant data extraction included the author(s), the year of publication, the type of study, the objective(s) of the study, the characteristics of the participants, the instruments, and the outcomes. The extracted results were presented and analysed using a “Data Charting” table [24]. This selection process took account of the PCC and is detailed in the flowchart PRISMA-ScR (Figure 1). The results extracted are described in relation to the questions outlined in the context of the study’s objective.

### 2.5. Analysis and Results Presentation

Results are summarised in tabular format, presenting systematically extracted data categories that address the study objectives and research questions. The data synthesis incorporates key characteristics from included studies to facilitate comprehensive analysis.

## 3. Results

### 3.1. Selection of Evidence Sources

The initial search strategy identified 171 potentially relevant studies (PubMed: 7 articles, Science Direct: 84 articles, PEDro: 6 articles, Cochrane: 10 articles, NIH: 23 articles, Scielo: 31 articles, ESBCO: 8 articles). After a search on Google Scholar, two additional studies were found. This was followed by the removal of eight duplicate studies. A total of 104 studies were excluded because they were not available in full text (PubMed: 1, Science Direct: 73, Cochrane: 5, NIH: 20, ESBCO: 2) and 3 because they were not published in English or Portuguese (Scielo: 2, EBSCO: 1). A total of 59 references were assessed by title and abstract. Subsequently, eight were excluded because they were not eligible in terms of the type of study (PubMed: 2, Science Direct: 1, PEDro: 3, Scielo: 2). A total of 51 articles were eligible for full-text evaluation. Of these, 47 studies did not meet the PCC inclusion criteria: 23 were excluded based on population (PubMed: 1; Science Direct: 3; PEDro: 1; Cochrane: 2; NIH: 2; Scielo: 12; EBSCO: 2), 21 based on concept (PubMed: 1; Science Direct: 4; PEDro: 1; Scielo: 14; EBSCO: 1), and 3 based on context (Science Direct: 1; PEDro: 1; Scielo: 1). Following this process, 4 studies were included in the scoping review: 2 from PubMed [11,25] and 2 from Science Direct and Google Scholar [25,26]. After screening, four studies were selected for this scoping review, identified through the databases PubMed (n = 2) [11,27], ScienceDirect, and Google Scholar (n = 2) [25,26], respectively.

After extracting the data, a narrative synthesis was carried out to describe the articles included in terms of the type of study, intervention and the type of participants included in the study. Table 2 shows the characteristics of eligible studies.

### 3.2. Types of Study

After the study selection process (Table 2), four studies were included in the present scoping review: three cross-sectional studies [11,25,26] and one cohort study [27]. Regarding the year of publication, these studies were published between 2015 and 2022, with all having been published within the last 10 years. All studies were published in English.

### 3.3. Participant Characteristics

The total sample across the four studies comprised 611 participants, all of whom were female. Two studies included exclusively gymnastics disciplines, namely 22 trampolinists [27] and 107 rhythmic gymnasts [25]. One study featured 9 artistic gymnasts and trampolinists within a group encompassing several sports: 23 volleyball players, 9 judokas, and 26 swimmers, in order to compare these disciplines with 96 non-athlete participants [11]. Another study included 68 artistic gymnasts, 116 team gymnasts, and 135 cheerleaders [26]. Overall, the athlete sample totalled 515 participants, with 322 gymnasts. The first study did not differentiate between the two gymnastics disciplines represented in the results (artistic gymnastics and trampoline) [11].

### 3.4. Intervention

Three studies [25,26,27] evaluated only gymnasts, all of whom were selected and assessed during national championships. However, Gram and Bø (2020) [25] also collected data in a training context. One study [11] compared the prevalence of urinary incontinence (UI) between two groups: athletes (from four different sports disciplines) and non-athletes. This study was the only one to include an informational session on the pelvic floor and its training prior to administering the questionnaire. All studies assessed participants using self-administered questionnaires. However, for supervision, Almeida et al. (2016) [11] made an investigator available, and Gram and Bø (2020) [25] provided a physiotherapist. All questionnaires, designed by the researchers, included demographic data and UI assessment. One study [25] conducted a clinical assessment of benign joint hypermobility. This involved evaluating passive extension of each fifth finger beyond 90 degrees, passive opposition of each thumb to the forearm, hyperextension of each elbow beyond 190 degrees, hyperextension of each knee beyond 10 degrees, and trunk flexion.

### 3.5. Assessment Tools

Regarding the assessment of urinary incontinence (UI), the International Consultation on Incontinence Questionnaire—Short Form (ICIQ-UI-SF or ICIQ-SF) was used in 100% of the studies. Questionnaires for the evaluation of anal incontinence were also identified [28,29]: the International Consultation on Incontinence Questionnaire Anal Incontinence Symptoms and Quality of Life Module (ICIQ-B) and the Fecal Incontinence Severity Index (FISI). For other symptoms, the following tools were observed: the Low Energy Availability in Females Questionnaire (LEAF-Q), the Low Energy Availability in Females Questionnaire—Short Form (LEAF-SF), the Triad-specific self-report questionnaire, and the Female Sexual Function Index (FSFI) for assessing the Female Athlete Triad [30,31]. The Beighton Score was used for assessing joint hypermobility [32], and the Rome III Criteria for functional gastrointestinal disorders (The Rome III diagnostic criteria for functional gastrointestinal disorders,” 2012). Vaginal and sexual symptoms [33] were also assessed through the International Consultation on Incontinence Questionnaire—Vaginal Symptoms (ICIQ-VS) and the Female Sexual Function Index (FSFI).

### 3.6. Outcomes

The primary outcome assessed in all studies was the prevalence of urinary incontinence (UI). Subsequently, the occurrence of each type of UI [25,26], anal incontinence (AI) [11,26], and pelvic organ prolapse [11] was evaluated. One study [11] assessed the influence of the sports discipline on the occurrence of UI, while another [27] examined the association between UI severity and training volume. Two studies [11,27] evaluated the impact of UI on quality of life, and one [27] also analysed its impact on athletic performance. Two other studies [25,26] investigated the impact of UI on athletic performance and outcomes. One study [11] analysed athletes’ attitudes towards UI, one [25] assessed knowledge about pelvic floor muscles (PFM), and one [26] examined PFM training and UI.

## 4. Discussion

The aim of this scoping review was to summarise the current evidence on urinary incontinence (UI) in adolescent and adult female gymnasts, to identify which gymnastics disciplines have the highest prevalence of UI (approximately 70%), and to determine which instruments and outcomes are used in the assessment and intervention of UI in athletes.

Characterisation

This study showed that the prevalence of UI is high in all the gymnastics disciplines assessed, with a higher prevalence of SUI. The modalities with the highest recorded prevalence were artistic gymnastics and trampolining, with a rate of 88.9 percent [11], and the lowest in rhythmic gymnastics, with a rate of 31.8 percent [25]. As for the type of UI, two studies [25,26] evaluated the different types of UI and found a higher prevalence for SUI. The other two studies [11,27] only reported the prevalence of SUI, since it was the most prevalent type of UI. There was a lack of physiotherapy intervention as a tool in the prevention and intervention of UI in the studies included in this scoping review.

Intervention

The main methods for intervening in UI are physiotherapy, biofeedback, pharmacology and, in more serious cases, surgery [34,35]. Physiotherapy works to promote continence through pelvic floor muscle (PFM) training programs. This method is recommended as a first-line intervention, with no reported adverse effects and a level of evidence classified as 1A, meaning strong evidence supported by consistent results from high-quality randomised controlled trials. It is particularly effective for stress urinary incontinence (SUI) in the general female population [36]. The National Institute for Health and Care Excellence (NICE) guideline indicates that PFM training is as effective as or more effective than surgery for around half of women with SUI [37]. Training should be orientated based on the deficits found and designed to develop specific function parameters for this muscle group. This is well documented as improving muscle strength, functionality, and blood supply to muscle tissue [20]. However, to date, there have been few studies on the effect of PFM training on elite athletes and gymnasts [13]. Due to the nature of the studies included, no specific interventions were identified, so this information has not been incorporated into Table 2.

Assessment tools/Outcomes

However, in terms of the use of assessment instruments, the most prevalent was the ‘International Consultation on Incontinence Questionnaire—Urinary Incontinence-Short Form’ (ICIQ-UI-SF). This specific and brief questionnaire aims to identify the presence of UI, its type and frequency, as well as its impact on the athletes’ quality of life [38,39]. This questionnaire is simple, quick to complete, has high levels of validity, reliability and sensitivity, and can be used in clinical practice and research [40]. The known risk factors for UI are pregnancy, eutocic labour, pelvic floor surgery, obesity, and age [5]. However, as Gram and Bø (2020) [25] highlight, there is a notable lack of research examining risk factors for urinary incontinence (UI) in young nulliparous women. While limited evidence suggests that a low body mass index (BMI), extensive hours of training, and joint hypermobility may be potential risk factors, the mechanisms by which these factors contribute to pelvic floor dysfunction remain poorly understood. Joint hypermobility, for instance, could theoretically increase susceptibility to UI by reducing the stability and support provided by connective tissues, including those of the pelvic floor. Similarly, high training volumes and low BMI may influence intra-abdominal pressure dynamics and tissue resilience, potentially exacerbating the risk of leakage during high-impact activities. Nevertheless, the assessment and analysis of these factors across studies are inconsistent, underscoring the need for further research to clarify their roles and interactions in the development of UI among young female athletes. The advantage of the ICIQ-UI-SF self-completion questionnaire also lies in its assessment of quality of life, an outcome that was considered in all the studies. This allows us to raise some hypotheses about the repercussions of this condition on young athletes. Studies indicate that athletes report a negative impact on quality of life [11] and sports performance [25,26]. Although one study revealed that female athletes reported only a “slight” impact of UI on their quality of life, the interpretation of these results should be performed with some caution, considering the sample limitation of only 22 athletes [27]. However, the authors suggest that shame and poor UI literacy may be additional factors contributing to the underreporting of actual symptoms [27]. Strategies to manage episodes of urinary incontinence (UI) have also been reported, including bladder emptying before training or competition [11] and the use of pads to manage the consequences of UI [26]. Additionally, studies highlight athletes’ limited knowledge about pelvic floor anatomy [25,26], although they express interest in receiving training aimed at preventing or managing UI [26]. This indicates that athletes are generally unaware of the available preventive and therapeutic options. A pioneering study also identified a positive association between athletic performance, training volume, and UI severity, as well as a strong positive correlation between years of training and the ICIQ-UI-SF score, frequency, and amount of leakage [27]. It is important to note that this association may be influenced by the fact that higher-performing gymnasts often perform more complex and high-impact skills, which generate greater intra-abdominal pressure, potentially increasing the risk of leakage regardless of pelvic floor muscle strength. This extremely important information raises the debate about the lack of consensus among studies as to whether high-impact physical exercise, such as gymnastics, is a risk or protective factor for UI. There are two explanatory hypotheses about the effect of exercise on the pelvic floor: the first is that exercise strengthens the pelvic floor and reduces the risk of UI, and the second is that physical activity weakens the pelvic floor, increasing the risk of UI [12]. The increased risk of urinary incontinence (UI) in high-impact sports is often attributed to the repeated and substantial increases in IAP [41], which may contribute to the high prevalence of UI observed in gymnastics, where tumbling landings and other acrobatic elements generate significant ground reaction forces [42]. Based on this premise, some studies suggest that the elevated pressure and impact forces progressively weaken the pelvic floor, whereas others propose that athletes may in fact develop strong pelvic floor muscles but still experience leakage when performing skills that generate extreme forces and pressures. This divergence of perspectives reflects the ongoing debate regarding the underlying mechanisms of the high prevalence of UI in gymnastics. Da Roza et al. (2015) [27], defend the hypothesis that urine loss is not due to a morphological change or fatigue of the PFM, but is a consequence of the altered muscular response to a mechanical stimulus. The authors state that it could be speculated that the high stress on the PFM could have a negative impact on the intrafusal fibres, reducing their ability to respond to stretching, which could result in a delay in contraction. Other authors, Gram et Bø (2020) [25], state that the high prevalence of UI seen in rhythmic gymnastics indicates that intense exercise does not protect against UI. For now, it can only be speculated whether high exposure to high-impact activities can cause UI or whether the condition is due to underlying genetic factors, such as a low position of the pelvic floor within the pelvis, a weakness of the PFM and/or connective tissue and/or a delayed neurophysiological response to increased intra-abdominal pressure. Finally, Skaug et al. (2022) [26], state that PFM fatigue during exercise could be another possible mechanism of SUI in gymnasts and cheerleaders. However, the fact that UI does not seem to bother them during daily activities indicates that the impact during training is not enough to induce incontinence and that the losses may be mainly related to sporting activities. This suggests that not only is there no consensus between the two hypotheses, but that high-impact sports such as gymnastics are a risk factor for the development of SUI in young nulliparous women. The studies emphasise the urgency of prioritising the prevention and management of urinary incontinence (UI) among both the athletes themselves and the professionals who support them, whether physiotherapists or other staff. Given that many gymnasts, particularly those at non-elite levels, may not have regular access to physiotherapists or specialised medical teams, it is important to promote pelvic floor health literacy among coaches and other relevant personnel. Educational strategies should be tailored to the level of training of these professionals, focusing on evidence-based, non-invasive approaches that can be communicated clearly without requiring specialised medical knowledge. For adolescent gymnasts, where the topic of pelvic floor anatomy may be sensitive, it is essential to involve parents or guardians in the discussion and to ensure that information is delivered in an age-appropriate and culturally sensitive manner. In this context, the physiotherapist’s role extends beyond rehabilitation, encompassing awareness-raising, communication, and education, with the aim of fostering empowerment, knowledge, and self-management strategies for UI prevention and intervention. This study has limitations. First, the scarcity of the available literature and the small number of included studies make it difficult to compare results and establish clear guidelines for interventions in this population. Second, most studies investigating urinary incontinence (UI) in athletes are observational, with very few evaluating specific interventions. Third, although the use of uniform assessment methods—predominantly questionnaires and other standardised instruments—improves comparability between studies, it may also limit the breadth of conclusions that can be drawn and reduce the robustness of the findings. These limitations highlight both the challenges faced in this field and the need for further research, which could inform the development of targeted and effective preventive programmes to improve pelvic floor health in gymnasts.

## 5. Conclusions

SUI was the most prevalent type of UI among female gymnasts, with the highest rates observed in artistic and trampoline disciplines. All included studies used the ICIQ-UI-SF to assess UI. Reported outcomes comprised the prevalence of UI and SUI, their impact on quality of life and athletic performance, and athletes’ knowledge about UI and the pelvic floor. None of the studies investigated or implemented any intervention strategies. Future research should prioritise the development of intervention studies, particularly experimental designs evaluating specific interventions, such as pelvic floor muscle training programs tailored to these athletes. It also seems important to increase athletes’ knowledge about UI prevention and management, reducing the associated stigma and encouraging a holistic approach to women’s health in sport.

## Figures and Tables

**Figure 1 sports-13-00319-f001:**
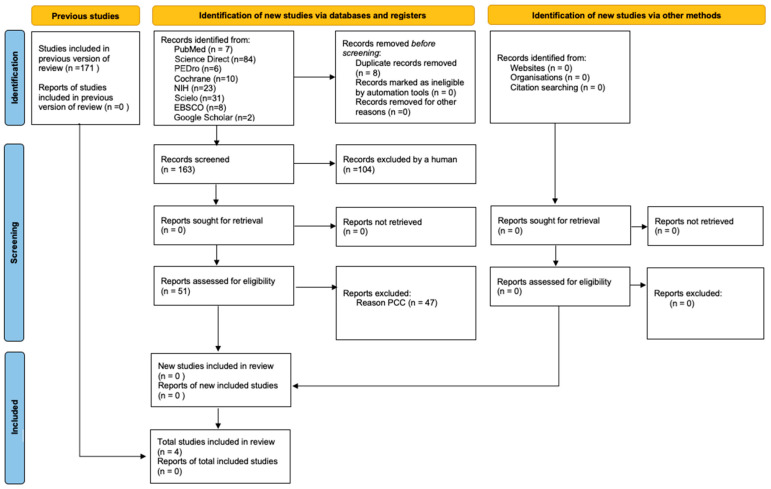
Flowchart of the scoping review according to the PRISMA-ScR model.

**Table 1 sports-13-00319-t001:** Search strategy according to database.

Database	Search Strategy	Filters Used
PubMed	(“woman” OR “female” OR “athlete”) AND (“gymnastic” OR “trampoline” OR “acrobatic” OR “high impact sport”) AND (“urinary incontinence” OR “stress urinary incontinence” OR “pelvic floor disorders” OR “loss of urine” OR “urine leakage”) AND (“prevalence” OR “treatment” OR “knowledge” OR “impact” OR “quality of life” OR “prevention”)	Published between 2012 and 2023 Full text available Study type: journal article, clinical trial, randomised controlled trial, books and documents Language: English, Portuguese Age: adolescents (13–18 years), young adults (19–24 years), adults (19–44 years) Sex: female
Cochrane	(“gymnastic” OR “trampolim” OR “acrobatic” OR “high impact sport”) AND (“urinary incontinence” OR “stress urinary incontinence” OR “pelvic floor disorders”)	Published between 2012 and 2023 Language: English Publication type: Clinical trials
Science Direct	(“gymnastic” OR “trampolim” OR “acrobatic” OR “high impact sport”) AND (“urinary incontinence” OR “stress urinary incontinence” OR “pelvic floor disorders”).	Published between 2012 and 2023 Publication type: research articles, book chapters Access: free access, open archive
Scielo	(“gymnastic” OR “trampolim” OR “acrobatic” OR “high impact sport”) AND (“urinary incontinence” OR “stress urinary incontinence” OR “pelvic floor disorders”)	Published between 2012 and 2023 Language: English, Portuguese Type of literature: articles
EBSCO	(“gymnastic” OR “trampolim” OR “acrobatic” OR “high impact sport”) AND (“urinary incontinence” OR “stress urinary incontinence” OR “pelvic floor disorders”)	Published between 2012 and 2023 References available Type of publication: academic journal articles, reports, books
PEDro	“incontinence. Subdisciplina: “sports”	Published from 2012 onwards Method: clinical trial
NIH	“urinary incontinence”. Outros termos: “sport”	Articles published from 2012 onwards Study type: clinical trial, observational study Status: completed and closed Expanded access: available Study with results Age: children and adults Sex: female

**Table 2 sports-13-00319-t002:** Characteristics of eligible studies.

Authors /Year	Study	Objectives	Participants	Assessment Tools	Outcomes
Almeida et al. (2016) [11]	Cross-sectional study	To investigate the occurrence of pelvic floor dysfunction (PFD) symptoms among athletes and non-athletes. To investigate the influence of sport on the occurrence and severity of urinary dysfunction.	n = 163 Athletes (n = 67): artistic gymnastics and trampoline (n = 9) Non-athletes (n = 96) 15–29 years BMI athletes: 21.7 (±2.6) BMI non-athletes: 20.9 (±3.9) Nulliparous IU (gymnasts): 88.9% IUE (gymnasts): 87.5%	ICIQ-UI-SF FISI Criteria Rome III FSFI ICIQ-VS	Pelvic floor dysfunctions Influence of modality Impact on quality of life Attitude towards UI
Da Roza et al. (2015) [27]	Cohort study	To investigate the association between UI severity and training volume and athletic performance in young female nulliparous trampolinists.	n = 22 Trampolinists/ National level 14–25 years BMI: 20.4 (±1.3) Nulliparous SUI: 72.7%	ICIQ-UI-SF	Prevalence of pelvic floor dysfunction (PFD) Association between UI severity and training volume Impact on quality of life and athletic performance
Gram et Bø, (2020) [25]	Cross-sectional study	To investigate the prevalence and risk factors of UI in rhythmic gymnasts and the impact of UI on sports performance. Evaluate PFM knowledge and PFM training.	n = 107 Rhythmic gymnastics International level 12–21 years BMI: 18.5 (±5.3) Nulliparous 65.4% menarche UI: 31.8% SUI: 61.8% UI: 8.8% UI: 17.6% Other UI: 11.8%	ICIQ-UI-SF “Triad-specific self-report questionnaire” LEAF-SF Beighton score	UI prevalence Prevalence of type of UI Impact of UI on athletic performance Knowledge about PFM and its training
Skaug et al., (2022) [26]	Cross-sectional study	To investigate the prevalence and risk factors of UI and anal incontinence (AI) in high-performance female artistic gymnasts (AG), team gymnasts (TG) and female cheerleaders. To investigate the impact of UI/IA on sports performance. To assess the athletes’ knowledge of PFM.	n = 319 Artistic gymnastics (n = 68), Team gymnastics (n = 116), Cheerleading (n = 135) National and international level 12–36 years BMI: 21.7 (±2.7) Nulliparous 92.2% menarcas IU: AG 70.6%, TG 83.6% IUE: AG 70.6%, TG 80.2% IUU: AG 8.8%, TG 12.9%	ICIQ-UI-SF ICIQ-B LEAF-Q	Prevalence of UI and AI Prevalence of type of UI Impact on athletic performance Knowledge of UI

AG—artistic gymnastics; TG—team gymnastics; AI—anal incontinence; ICIQ-B—International Consultation on Incontinence Questionnaire—Anal Incontinence Symptoms and Quality of Life Module; ICIQ-UI-SF—International Consultation on Incontinence Questionnaire-Short Form; BMI—body mass index; UI—urinary incontinence; SUI—stress urinary incontinence; e; LEAF-Q—Low Energy Availability in Females Questionnaire—Short Form; PFM—pelvic floor muscles.

## Data Availability

No new data were created or analyzed in this study. Data sharing is not applicable to this article.

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
