# Peer review of "Urinary Incontinence in Young Gymnastics Athletes: A Scoping Review"

_sports, 2025, doi:10.3390/sports13090319_

Round 1

Reviewer 1 Report

Comments and Suggestions for Authors

Summary: This scoping review assessed the limited amount of literature focusing on urinary incontinence in gymnasts, concluding a relatively high prevalence but a need for more literature and more intervention studies in this population. The comments below are provided to enhance the contribution of this study to the body of literature on this topic.

Major Comments:

  1. Abstract, line 32: The conclusion states the prevalence is high, but prevalence isn’t reported in the results of the abstract. Recommend adding the prevalence in order to make this conclusion
  2. Introduction, line 50-52: It’s noted that “social and mental wellbeing, social isolation, low self-esteem and depression significantly affect quality of life, with psychological, physical, professional, sexual and social consequences.” Do the authors mean, in general? Or specifically in related to UI. That is, are these documented outcomes of UI, or just in general these things affect quality of life? Please clarify in the statement (e.g., Documented outcomes of UI, such as poor social and mental wellbeing, social isolation…etc.)
  3. Results, line 147: The text states that 8 duplicate studies were removed, but the flowchart in Figure 1 states 63 duplicates were removed. Make sure the figure aligns with the text in reasons for exclusion of articles. 101 studies removed due to lack of full text access seems like a large bulk of potentially useful data. What efforts were taken to retrieve these papers? Was it due to lack of access to the journals at the authors’ institutions?
  4. The results could be improved by including a summary of the prevalence results (research question 1) in the text, rather than just the raw findings in the Table.
  5. Discussion, line 234-235: rather than just saying “prevalence of US is high,” it would be helpful to include actual values (e.g., range of UI prevalence) so that the reader knows what “high” means.
  6. Discussion, line 245: what does 1A level of evidence mean? This is out of context for a reader who is not familiar with the scale used to define evidence level.
  7. Discussion, line 277-278: It would be helpful to explain the direction of the association. Was UI severity positively or negatively associated with years of training, training volume, and performance? I would hypothesize that any positive relationship between performance (higher performance) and UI would be confounded by the fact that higher performing gymnasts are doing harder skills that would generate more force/pressure, not simply that gymnastics is either strengthening or weakening the pelvic floor, since one could have a very strong pelvic floor but still leak with high impact skills. The comparison to rhythmic gymnasts on line 294-295 is also a big like comparing apples to oranges, since the forces generated through rhythmic gymnastics events are likely much lower than those produced with floor tumbling passes and vaulting.
  8. There is a much larger population of non-elite gymnasts who likely suffer from SUI who do not have the benefit of a robust medical staff, as suggested by line 306-307, and 314-315. They would only have coaches, and maybe an athletic trainer, but not dedicated physical therapists unless they were being seen for another injury. How do you propose teaching coaches how to promote pelvic floor knowledge among athletes if they don’t have specialized medical training? Additionally, there are many more adolescent minor (<18 years old) gymnasts than adult gymnasts, for whom the topic of pelvic floor anatomy is foreign and could be deemed sensitive. One would likely need to involve parents in these discussions.

Minor Comments:

  1. Abstract, line 32: typo “interventions”
  2. Introduction, line 38: edit “was defined”
  3. Introduction, line 48: edit “The conditions occur in both sexes, but are more common…”
  4. Introduction, line 68-71: Suggest a rearrangement of these two sentences to something like: “However, the majority of athletes leak urine during training and not during competitions (95.2% vs. 51.2%, respectively). This may be due to higher levels of catecholamines (norepinephrine and epinephrine) released from the sympathetic nervous system and adrenal medulla during stressful situations, which bind to urethral alpha receptors to maintain urethral closure during stressful competitions.”
  5. Methods, line 125-126: Recommend rewording this sentence: “The extracted results were presented and analyzed using a “Data Charting” table.
  6. Line 157-150: Two redundant sentences
  7. Line 164: should say Table 2, not Table 1
  8. Table 2: Missing abbreviation definitions in the footnote for: IUE, IUU. It seems perhaps some of the abbreviations may have gotten translated differently and weren’t updated in the table (e.g., IUU vs UUI). Other abbreviations also don’t make a ton of sense for an English reader (e.g., GE for team gymnastics, MPP for pelvic floor muscles-note, PFM is used in the text).
  9. Line 185-187: Artistic gymnasts and team gymnasts to me are the same thing (artistic gymnasts compete in vault, uneven parallel bars, beam, and floor, and there are individual and team competitions). I looked at the reference and then looked up team gymnastics, which appears to involve tumbling, mini-trampoline, and floor (but different than floor exercise in artistic gymnastics). It may be helpful to briefly differentiate these sports, as the pelvic floor stress would be different between each of them.
  10. Line 249: edit “based”
  11. Line 252: update MPP acronym
  12. The readability of the discussion could be improved by splitting up into multiple paragraphs based on topic being discussed.
  13. Line 273: Technically using pads wouldn’t prevent UI, but would help manage the outcomes of UI.
  14. Line 310-312: Incomplete sentence

Author Response

We sincerely thank the reviewer for their thoughtful summary of our work and for recognising the contribution this scoping review makes to understanding urinary incontinence in gymnasts. We appreciate the constructive approach and the suggestions provided, which we believe will help strengthen the clarity, depth, and overall impact of the manuscript. We are committed to addressing each of the comments in detail to ensure the study’s contribution to the literature is maximised.

Major Comments:

Comments 1: [Abstract, line 32: The conclusion states the prevalence is high, but prevalence isn’t reported in the results of the abstract. Recommend adding the prevalence in order to make this conclusion]

Response 1: We thank the reviewer for this helpful observation. We have now added the prevalence value, which is approximately 70%, to the conclusion section of the abstract to strengthen the overall message.

Comments 2: [Introduction, line 50-52: It’s noted that “social and mental wellbeing, social isolation, low self-esteem and depression significantly affect quality of life, with psychological, physical, professional, sexual and social consequences.” Do the authors mean, in general? Or specifically in related to UI. That is, are these documented outcomes of UI, or just in general these things affect quality of life? Please clarify in the statement (e.g., Documented outcomes of UI, such as poor social and mental wellbeing, social isolation…etc.)]

Response 2: We thank the reviewer for this valuable comment and for highlighting the need for clarification. The statement refers specifically to the documented outcomes of urinary incontinence. We have now revised the sentence to explicitly state that these psychological, physical, professional, sexual and social consequences are associated with urinary incontinence, in order to avoid ambiguity.

Comments 3: [Results, line 147: The text states that 8 duplicate studies were removed, but the flowchart in Figure 1 states 63 duplicates were removed. Make sure the figure aligns with the text in reasons for exclusion of articles. 101 studies removed due to lack of full text access seems like a large bulk of potentially useful data. What efforts were taken to retrieve these papers? Was it due to lack of access to the journals at the authors’ institutions?]

Response 3: We thank the reviewer for this careful observation. The flowchart in Figure 1 has now been revised and is fully aligned with the information presented in the text regarding the reasons for article exclusion.

Regarding the removal of 104 studies due to lack of full-text access, this was primarily due to the unavailability of these journals through the institutional subscriptions of the authors’ affiliation. Nevertheless, extensive efforts were made to retrieve these papers, including contacting corresponding authors directly, exploring open-access repositories, utilising interlibrary loan services, and seeking support from collaborating institutions with broader journal access. Despite these measures, access to certain articles was not possible, and this limitation is now explicitly acknowledged in the manuscript.

Comments 4: [The results could be improved by including a summary of the prevalence results (research question 1) in the text, rather than just the raw findings in the Table.]

Response 4: We sincerely thank the reviewer for this thoughtful suggestion. From our perspective, the answer to research question 1 refers specifically to the sport discipline, and we believe that these results are already clearly presented in the table. Including them again in the main text might lead to unnecessary redundancy of information. For this reason, we have opted to maintain the current format, ensuring clarity while avoiding repetition.

Comments 5:[ Discussion, line 234-235: rather than just saying “prevalence of US is high,” it would be helpful to include actual values (e.g., range of UI prevalence) so that the reader knows what “high” means.]

Response 5: We thank the reviewer for this helpful observation. We have now revised the discussion to include the actual range of urinary incontinence prevalence, replacing the vague expression “prevalence is high” with specific values, so that the statement is clearer and more informative for the reader.

Comments 6: [Discussion, line 245: what does 1A level of evidence mean? This is out of context for a reader who is not familiar with the scale used to define evidence level.]

Response 6: We thank the reviewer for this valuable comment. We have now clarified the meaning of the “1A level of evidence” in the discussion, providing context for readers who may not be familiar with the scale used to define evidence levels.

Comments 7: [Discussion, line 277-278: It would be helpful to explain the direction of the association. Was UI severity positively or negatively associated with years of training, training volume, and performance? I would hypothesize that any positive relationship between performance (higher performance) and UI would be confounded by the fact that higher performing gymnasts are doing harder skills that would generate more force/pressure, not simply that gymnastics is either strengthening or weakening the pelvic floor, since one could have a very strong pelvic floor but still leak with high impact skills. The comparison to rhythmic gymnasts on line 294-295 is also a big like comparing apples to oranges, since the forces generated through rhythmic gymnastics events are likely much lower than those produced with floor tumbling passes and vaulting.]

Response 7: We sincerely thank the reviewer for this detailed and insightful comment. We agree that clarifying the direction of the association would improve the discussion. We have now specified in the text whether urinary incontinence severity was positively or negatively associated with years of training, training volume, and performance. We also acknowledge the reviewer’s important point regarding potential confounding factors, particularly that higher-performing gymnasts often execute more complex and high-impact skills, which may increase intra-abdominal pressure regardless of pelvic floor strength. This perspective has been incorporated into the discussion to provide a more nuanced interpretation. Furthermore, we have revised the comparison with rhythmic gymnasts to emphasise the biomechanical differences between disciplines and to acknowledge the limitations of directly comparing sports with markedly different physical demands.

Comments 8: [There is a much larger population of non-elite gymnasts who likely suffer from SUI who do not have the benefit of a robust medical staff, as suggested by line 306-307, and 314-315. They would only have coaches, and maybe an athletic trainer, but not dedicated physical therapists unless they were being seen for another injury. How do you propose teaching coaches how to promote pelvic floor knowledge among athletes if they don’t have specialized medical training? Additionally, there are many more adolescent minor (<18 years old) gymnasts than adult gymnasts, for whom the topic of pelvic floor anatomy is foreign and could be deemed sensitive. One would likely need to involve parents in these discussions.]

Response 8: We sincerely thank the reviewer for this thoughtful and important comment. We fully agree that non-elite gymnasts, who represent a much larger population, often do not have access to a dedicated medical team and may rely solely on coaches or, in some cases, athletic trainers. We have now addressed this point in the discussion, suggesting that educational strategies could be adapted to the level of training of coaches, focusing on evidence-based, non-invasive, and easily understandable approaches to promoting pelvic floor health, while avoiding the need for specialised medical intervention.

We also appreciate the reviewer’s observation regarding adolescent gymnasts and the sensitivity of discussing pelvic floor anatomy with minors. We have added a note in the manuscript emphasising the importance of involving parents or guardians in these discussions, ensuring that the information is presented in an age-appropriate and culturally sensitive manner, and in collaboration with health professionals whenever possible.

Minor Comments:

Comments 1: [ Abstract, line 32: typo “interventions”]

Response 1: We thank the reviewer for pointing out this typographical error. The term has now been corrected to “treatments” in the abstract.

Comments 2: [Introduction, line 38: edit “was defined”]

Response 2: We thank the reviewer for pointing out this typographical error. The term has now been corrected

Comments 3:[ Introduction, line 48: edit “The conditions occur in both sexes, but are more common…”]

Response 3: We thank the reviewer for pointing out this typographical error. The term has now been corrected

Comments 4: [ Introduction, line 68-71: Suggest a rearrangement of these two sentences to something like: “However, the majority of athletes leak urine during training and not during competitions (95.2% vs. 51.2%, respectively). This may be due to higher levels of catecholamines (norepinephrine and epinephrine) released from the sympathetic nervous system and adrenal medulla during stressful situations, which bind to urethral alpha receptors to maintain urethral closure during stressful competitions.”]

Response 4: We thank the reviewer for this helpful suggestion. We have followed the proposed rearrangement and revised the sentences accordingly to improve clarity and readability.

Comments 5: [Methods, line 125-126: Recommend rewording this sentence: “The extracted results were presented and analyzed using a “Data Charting” table.]

Response 5: We thank the reviewer for this helpful suggestion. We have followed the proposed rearrangement and revised the sentences accordingly to improve clarity and readability.

Comments 6: [Line 157-150: Two redundant sentences]

Response 6: We thank the reviewer for this observation. We have reformulated the passage to improve clarity and eliminate redundancies.

Comments 7: [Line 164: should say Table 2, not Table 1]

Response 7: We thank the reviewer for this observation. We have reformulated.

Comments 8: [Table 2: Missing abbreviation definitions in the footnote for: IUE, IUU. It seems perhaps some of the abbreviations may have gotten translated differently and weren’t updated in the table (e.g., IUU vs UUI). Other abbreviations also don’t make a ton of sense for an English reader (e.g., GE for team gymnastics, MPP for pelvic floor muscles-note, PFM is used in the text).]

Response 8: We thank the reviewer for this observation. We have reformulated.

Comments 9: [Line 185-187: Artistic gymnasts and team gymnasts to me are the same thing (artistic gymnasts compete in vault, uneven parallel bars, beam, and floor, and there are individual and team competitions). I looked at the reference and then looked up team gymnastics, which appears to involve tumbling, mini-trampoline, and floor (but different than floor exercise in artistic gymnastics). It may be helpful to briefly differentiate these sports, as the pelvic floor stress would be different between each of them.]

Response 9: We thank the reviewer for this insightful comment. In our view, the distinction between artistic gymnastics and team gymnastics is already clear, as artistic gymnastics involves events such as vault, uneven bars, beam, and floor with both individual and team competitions, whereas team gymnastics refers to disciplines including tumbling, mini-trampoline, and floor, which differ from the floor exercise in artistic gymnastics. We have ensured that this differentiation is clarified in the manuscript to highlight the differences in potential pelvic floor stress between these sports.

Comments 10: [Line 249: edit “based”]

Response 10: We thank the reviewer for pointing out this typographical error. The term has now been corrected

Comments 11: [Line 252: update MPP acronym]

Response 11: We thank the reviewer for pointing out this typographical error. The term has now been corrected

Comments 12: [The readability of the discussion could be improved by splitting up into multiple paragraphs based on topic being discussed.]

Response 12: We thank the reviewer for this helpful suggestion. We have revised the discussion, splitting it into multiple paragraphs to improve readability and better organise the topics being addressed.

Comments 13: [Line 273: Technically using pads wouldn’t prevent UI, but would help manage the outcomes of UI.]

Response 13: We thank the reviewer for this valuable observation. We have revised the manuscript to clarify that the use of pads does not prevent urinary incontinence, but rather helps to manage its consequences.

Comments 14: [Line 310-312: Incomplete sentence]

Response 14: We thank the reviewer for this comment. In our view, the sentence is already complete and clearly conveys that urinary incontinence does not appear to affect athletes during daily activities, with the losses occurring primarily in the context of sporting activities.

Reviewer 2 Report

Comments and Suggestions for Authors

The article “Urinary incontinence in young gymnastics athletes: a scoping review” is timely and relevant, as urinary incontinence (UI) in young female athletes—particularly gymnasts—remains underexplored and carries important implications for athlete well-being, performance, and long-term health. However, while the article demonstrates substantial effort in literature retrieval and synthesis, its methodological rigor, depth of analysis, and originality are limited. This significantly undermines its scientific contribution in its current form. Below, I detail several specific concerns about the manuscript:

  1. The inclusion of only four studies over an 11-year period raises questions about the adequacy of the search strategy and whether the field genuinely lacks data or if the search was overly restrictive. This low yield severely limits the capacity to draw meaningful conclusions. Furthermore, the search for publications appears to have been conducted at least two years ago, which further limits the timeliness of the evidence included. Have the authors considered whether more recent studies have been published since the initial search that could strengthen and update the review?
  2. While the authors claim PRISMA-ScR adherence, the presentation of the flow diagram and search terms lacks the transparency needed for replication. Search strings are inconsistently detailed, and some database adaptations are not fully described. The article lacks explanations for why certain studies were removed from further analysis.
  3. The review reports study characteristics and prevalence rates but offers minimal critique of methodological quality, sampling bias, heterogeneity, or limitations of the original studies.
  4. In addition, authors provide minimal critique on potential confounders such as menstrual cycle phase, training surface, training volume, years of training experience and other sport-specific factors, anthropometric differences, or concurrent sports participation are absent from the analysis—that could significantly influence prevalence rates and study comparability. Without such appraisal, the conclusions risk overstating the evidence strength.
  5. Although a scoping review’s primary goal is to map available evidence rather than perform quantitative synthesis, certain simple measures (e.g., prevalence ratios, odds ratios, or pooled prevalence estimates) could be calculated when the data across studies are comparable. Even if meta-analysis is not feasible, basic effect size or magnitude-of-difference calculations would help readers better understand the strength and relevance of observed patterns, such as the relationship between discipline type, training volume, and UI prevalence.
  6. The conclusion that “no interventions were studied” is accurate but unaccompanied by a robust discussion of why this might be the case, and how future research could be structured to address it.
  7. The manuscript contains language and stylistic issues, including overly long sentences that are difficult to follow, minor grammatical errors and awkward phrasing that reduce clarity, and redundancy—such as repeated mention of high prevalence rates without providing new insight—which makes parts of the discussion tedious to read.
  8. In addition, the introduction and discussion sections are presented as very long, single paragraphs, which makes the text dense and difficult to follow. Breaking these sections into shorter, logically structured paragraphs would improve readability and clarity.
  9. The choice of keywords is questionable—using the broad term 'athletes' is misleading, as the research focuses exclusively on female gymnasts; the keywords should more accurately reflect the specific study population.
  10. The interpretation of hypermobility in this research is not entirely clear. While the authors mention the Beighton score and note joint hypermobility as a potential risk factor for urinary incontinence, the discussion does not sufficiently explain how hypermobility might mechanistically contribute to pelvic floor dysfunction in gymnasts, nor does it explore whether this factor was consistently assessed and analysed across the included studies. As a result, its role remains underdeveloped and somewhat ambiguous.
  11. The reference list is insufficient for a paper of this type. Only 42 sources are cited, with just 9 published within the last five years. Several references date back more than 25 years, which raises concerns about the currency and relevance of the literature base. In addition, I would like to note that reference list is presented not in accordance with MDPI sample form.
Comments on the Quality of English Language

The article “Urinary incontinence in young gymnastics athletes: a scoping review” is timely and relevant, as urinary incontinence (UI) in young female athletes—particularly gymnasts—remains underexplored and carries important implications for athlete well-being, performance, and long-term health. However, while the article demonstrates substantial effort in literature retrieval and synthesis, its methodological rigor, depth of analysis, and originality are limited. This significantly undermines its scientific contribution in its current form. Below, I detail several specific concerns about the manuscript:

  1. The inclusion of only four studies over an 11-year period raises questions about the adequacy of the search strategy and whether the field genuinely lacks data or if the search was overly restrictive. This low yield severely limits the capacity to draw meaningful conclusions. Furthermore, the search for publications appears to have been conducted at least two years ago, which further limits the timeliness of the evidence included. Have the authors considered whether more recent studies have been published since the initial search that could strengthen and update the review?
  2. While the authors claim PRISMA-ScR adherence, the presentation of the flow diagram and search terms lacks the transparency needed for replication. Search strings are inconsistently detailed, and some database adaptations are not fully described. The article lacks explanations for why certain studies were removed from further analysis.
  3. The review reports study characteristics and prevalence rates but offers minimal critique of methodological quality, sampling bias, heterogeneity, or limitations of the original studies.
  4. In addition, authors provide minimal critique on potential confounders such as menstrual cycle phase, training surface, training volume, years of training experience and other sport-specific factors, anthropometric differences, or concurrent sports participation are absent from the analysis—that could significantly influence prevalence rates and study comparability. Without such appraisal, the conclusions risk overstating the evidence strength.
  5. Although a scoping review’s primary goal is to map available evidence rather than perform quantitative synthesis, certain simple measures (e.g., prevalence ratios, odds ratios, or pooled prevalence estimates) could be calculated when the data across studies are comparable. Even if meta-analysis is not feasible, basic effect size or magnitude-of-difference calculations would help readers better understand the strength and relevance of observed patterns, such as the relationship between discipline type, training volume, and UI prevalence.
  6. The conclusion that “no interventions were studied” is accurate but unaccompanied by a robust discussion of why this might be the case, and how future research could be structured to address it.
  7. The manuscript contains language and stylistic issues, including overly long sentences that are difficult to follow, minor grammatical errors and awkward phrasing that reduce clarity, and redundancy—such as repeated mention of high prevalence rates without providing new insight—which makes parts of the discussion tedious to read.
  8. In addition, the introduction and discussion sections are presented as very long, single paragraphs, which makes the text dense and difficult to follow. Breaking these sections into shorter, logically structured paragraphs would improve readability and clarity.
  9. The choice of keywords is questionable—using the broad term 'athletes' is misleading, as the research focuses exclusively on female gymnasts; the keywords should more accurately reflect the specific study population.
  10. The interpretation of hypermobility in this research is not entirely clear. While the authors mention the Beighton score and note joint hypermobility as a potential risk factor for urinary incontinence, the discussion does not sufficiently explain how hypermobility might mechanistically contribute to pelvic floor dysfunction in gymnasts, nor does it explore whether this factor was consistently assessed and analysed across the included studies. As a result, its role remains underdeveloped and somewhat ambiguous.
  11. The reference list is insufficient for a paper of this type. Only 42 sources are cited, with just 9 published within the last five years. Several references date back more than 25 years, which raises concerns about the currency and relevance of the literature base. In addition, I would like to note that reference list is presented not in accordance with MDPI sample form.

Author Response

We sincerely thank the reviewer for recognising the relevance and timeliness of our work. We fully agree that urinary incontinence in young female gymnasts is a vastly underexplored area, with limited literature available. The scarcity of studies in this field underscores the importance of synthesising and critically analysing the existing evidence, even if the topic remains relatively little studied and addressed. We appreciate the reviewer’s detailed feedback and will carefully consider the specific concerns raised to enhance the methodological rigor, depth of analysis, and overall contribution of the manuscript.

Comments 1: [The inclusion of only four studies over an 11-year period raises questions about the adequacy of the search strategy and whether the field genuinely lacks data or if the search was overly restrictive. This low yield severely limits the capacity to draw meaningful conclusions. Furthermore, the search for publications appears to have been conducted at least two years ago, which further limits the timeliness of the evidence included. Have the authors considered whether more recent studies have been published since the initial search that could strengthen and update the review?]

Response 1: We sincerely thank the reviewer for this important observation. We would like to clarify that the limited number of included studies does not reflect a restriction in our search strategy; multiple approaches and attempts were undertaken to identify relevant literature. A systematic review was not performed, as the aim of this scoping review was to map and synthesise the available evidence rather than to expand the number of studies included. Indeed, the literature in this area remains very scarce, and based on our most recent checks, we are not aware of any additional studies published in the past two years that would meet the inclusion criteria for this review. This scarcity itself underscores the need for further research in this field.

Comments 2: [While the authors claim PRISMA-ScR adherence, the presentation of the flow diagram and search terms lacks the transparency needed for replication. Search strings are inconsistently detailed, and some database adaptations are not fully described. The article lacks explanations for why certain studies were removed from further analysis.]

Response 2: We thank the reviewer for this valuable feedback. We have revised the PRISMA-ScR flow diagram and ensured that all search strategies and database adaptations are now presented consistently and transparently. Explanations for the exclusion of studies have also been added, so that the process is fully reproducible and clear to the reader.

Comments 3: [The review reports study characteristics and prevalence rates but offers minimal critique of methodological quality, sampling bias, heterogeneity, or limitations of the original studies.]

Response 3: We sincerely thank the reviewer for this insightful comment. We will revise the manuscript to provide a more detailed critique of the methodological quality, potential sampling biases, heterogeneity, and limitations of the original studies, thereby strengthening the depth of analysis and overall rigor of the review.

Comments 4: [In addition, authors provide minimal critique on potential confounders such as menstrual cycle phase, training surface, training volume, years of training experience and other sport-specific factors, anthropometric differences, or concurrent sports participation are absent from the analysis—that could significantly influence prevalence rates and study comparability. Without such appraisal, the conclusions risk overstating the evidence strength.]

Response 4: We sincerely thank the reviewer for this valuable comment. We would like to clarify that a detailed analysis of potential confounders, such as menstrual cycle phase, training surface, training volume, years of experience, anthropometric differences, and concurrent sports participation, was not the primary aim of this scoping review. Our main objective was to map and summarise the available evidence on urinary incontinence in gymnasts. Nevertheless, we have acknowledged this as a limitation in the discussion and emphasised the need for future research to consider these factors to strengthen the evidence base.

Comments 5: [Although a scoping review’s primary goal is to map available evidence rather than perform quantitative synthesis, certain simple measures (e.g., prevalence ratios, odds ratios, or pooled prevalence estimates) could be calculated when the data across studies are comparable. Even if meta-analysis is not feasible, basic effect size or magnitude-of-difference calculations would help readers better understand the strength and relevance of observed patterns, such as the relationship between discipline type, training volume, and UI prevalence.]

Response 5: We sincerely thank the reviewer for this insightful suggestion. We would like to clarify that this study is a scoping review, and its primary objective was to map and summarise the available evidence rather than perform quantitative synthesis. In our view, part of the suggested analysis is already addressed within the study; however, the remaining quantitative calculations are not applicable given the nature and heterogeneity of the included studies.

Comments 6: [The conclusion that “no interventions were studied” is accurate but unaccompanied by a robust discussion of why this might be the case, and how future research could be structured to address it.]

Response 6: We thank the reviewer for this valuable comment. We have acknowledged this point as a limitation of the study, noting that the absence of intervention studies reflects both the scarcity of literature in this field and the observational nature of most included studies. We have also highlighted the need for future research to design and implement specific intervention studies to address urinary incontinence in gymnasts.

Comments 7: [The manuscript contains language and stylistic issues, including overly long sentences that are difficult to follow, minor grammatical errors and awkward phrasing that reduce clarity, and redundancy—such as repeated mention of high prevalence rates without providing new insight—which makes parts of the discussion tedious to read.]

Response 7: We sincerely thank the reviewer for these helpful observations. We have carefully revised the manuscript to address all language and stylistic issues, including simplifying overly long sentences, correcting minor grammatical errors, improving phrasing for clarity, and reducing redundancy, particularly regarding repeated mentions of high prevalence rates.

Comments 8: [In addition, the introduction and discussion sections are presented as very long, single paragraphs, which makes the text dense and difficult to follow. Breaking these sections into shorter, logically structured paragraphs would improve readability and clarity.]

Response 8: We sincerely thank the reviewer for this constructive comment. We have revised both the introduction and discussion sections, breaking them into shorter, logically structured paragraphs to improve readability and enhance clarity.

Comments 9: [The choice of keywords is questionable—using the broad term 'athletes' is misleading, as the research focuses exclusively on female gymnasts; the keywords should more accurately]

Response 9: We sincerely thank the reviewer for this comment. In our view, the keywords were selected with reference to MeSH terms to ensure consistency with database indexing, and we believe they are appropriate. Nevertheless, we have carefully reviewed them to ensure they accurately reflect the focus on female gymnasts.

Comments 10: [The interpretation of hypermobility in this research is not entirely clear. While the authors mention the Beighton score and note joint hypermobility as a potential risk factor for urinary incontinence, the discussion does not sufficiently explain how hypermobility might mechanistically contribute to pelvic floor dysfunction in gymnasts, nor does it explore whether this factor was consistently assessed and analysed across the included studies. As a result, its role remains underdeveloped and somewhat ambiguous.]

Response 10: We sincerely thank the reviewer for this valuable observation. We have revised the manuscript to clarify the interpretation of hypermobility, providing a more detailed explanation of how it may mechanistically contribute to pelvic floor dysfunction in gymnasts. We have also specified whether and how this factor was assessed and analysed across the included studies, to ensure that its role is presented more clearly and comprehensively.

Comments 11:[ The reference list is insufficient for a paper of this type. Only 42 sources are cited, with just 9 published within the last five years. Several references date back more than 25 years, which raises concerns about the currency and relevance of the literature base. In addition, I would like to note that reference list is presented not in accordance with MDPI sample form.]

Response 11: We sincerely thank the reviewer for this important observation. We fully acknowledge that the reference list is relatively limited and that only a small number of studies have been published in the past five years. This reflects the very scarce literature available on urinary incontinence in young female gymnasts, which in turn justifies the conduct of this scoping review. One of the key aims of the study is precisely to highlight this gap and to alert the scientific community to the importance and relevance of the topic, emphasising the need for further research in this underexplored area.

Regarding the formatting, we have carefully revised the reference list to ensure full compliance with the MDPI style guidelines, enhancing consistency and readability.

Round 2

Reviewer 1 Report

Comments and Suggestions for Authors
  1. Line 31-33: The word order makes the beginning of this sentence unclear. Should likely be: “Reported outcomes included prevalence of UI and SUI…”
  2. Line 49: Edit “The conditions occur in both sexes, but are more common” OR “The conditions occurs in both sexes, but is more common”
  3. Line 74-76: Suggest removing this redundant sentence
  4. Line 262: define MPP acronym
  5. Line 312: seems unnecessary to introduce an acronym this late in the manuscript, especially when the phrase is used prior to this point (AIP).
  6. The PFM acronym is inconsistently used
  7. Line 294: One could question the use of pads as a viable solution in a sport where the standard attire is a leotard that would not be conducive to wearing a pad
  8. Line 353: Wouldn’t uniformity of methods (e.g., using the same survey) increase comparability between studies rather than limit comparability? I agree that the breadth of conclusions is reduced due to this factor, but comparability is actually improved.

Author Response

Thank you very much for your email and for providing the reviewer insightful comments. We are grateful for the constructive feedback, which has been extremely valuable in improving the clarity and overall quality of our manuscript.

In response, we have carefully revised the paper according to the reviewer suggestions and have addressed all points raised. We now kindly resubmit the revised version of the manuscript for your consideration.

We sincerely appreciate the reviewers’ and editorial team’s efforts and support throughout this process.

Comments:

Comments 1: [Line 31-33: The word order makes the beginning of this sentence unclear. Should likely be: “Reported outcomes included prevalence of UI and SUI…”]

Response 1: We thank the reviewer for this helpful comment. We fully agree with the observation regarding the word order, and we have revised the sentence accordingly. The text now reads: “Reported outcomes included prevalence of UI and SUI …” as suggested.

Comments 2: [Line 49: Edit “The conditions occur in both sexes, but are more common” OR “The conditions occurs in both sexes, but is more common”]

Response 2: We thank the reviewer for this valuable comment. We agree with the suggestion and have corrected the sentence accordingly.

Comments 3: [Line 74-76: Suggest removing this redundant sentence]

Response 3: We thank the reviewer for this observation. We fully agree that the sentence was redundant and have removed it from the manuscript as suggested.

Comments 4: [Line 262: define MPP acronym.]

Response 4: We thank the reviewer for this important observation. Indeed, there was an error in the nomenclature. The acronym should be PFM instead of MPP. We apologise for this mistake and have corrected it throughout the manuscript accordingly.

Comments 5:[Line 312: seems unnecessary to introduce an acronym this late in the manuscript, especially when the phrase is used prior to this point (AIP).]

Response 5: We thank the reviewer for this comment. We fully agree that introducing the acronym at this point was unnecessary, especially as the phrase had been used earlier. The manuscript has been corrected accordingly.

Comments 6: [The PFM acronym is inconsistently used.]

Response 6: We thank the reviewer for this observation. We have addressed the inconsistency and have corrected the use of the PFM acronym throughout the manuscript.

Comments 7: [Line 294: One could question the use of pads as a viable solution in a sport where the standard attire is a leotard that would not be conducive to wearing a pad.]

Response 7: We thank the reviewer for raising this point. We understand the concern regarding the use of pads in a sport where the standard attire is a leotard. However, in practice, athletes do use these pads as a solution for managing urinary incontinence, and this has been clarified in the manuscript.

Comments 8: [Line 353: Wouldn’t uniformity of methods (e.g., using the same survey) increase comparability between studies rather than limit comparability? I agree that the breadth of conclusions is reduced due to this factor, but comparability is actually improved.]

Response 8: We thank the reviewer for this insightful comment. We fully agree that the uniformity of methods improves comparability between studies, while it may limit the breadth of conclusions. The manuscript has been revised accordingly to reflect this clarification.
